# Culture Isolate of *Rickettsia felis* from a Tick

**DOI:** 10.3390/ijerph19074321

**Published:** 2022-04-04

**Authors:** Monika Danchenko, Oldřich Benada, Ľudovít Škultéty, Zuzana Sekeyová

**Affiliations:** 1Department of Rickettsiology, Biomedical Research Center, Slovak Academy of Sciences, Dúbravská cesta 9, 845 05 Bratislava, Slovakia; monika.bohacsova@savba.sk (M.D.); ludovit.skultety@savba.sk (Ľ.Š.); 2Laboratory of Molecular Structure Characterization, Institute of Microbiology, Czech Academy of Sciences, Vídeňská 1038, 142 20 Prague, Czech Republic; benada@biomed.cas.cz

**Keywords:** *Rickettsia felis*, vector-borne bacteria, *Ixodes ricinus*, shell-vial technique, cell culture

## Abstract

Although the cat flea, *Ctenocephalides felis*, has been identified as the primary vector of *Rickettsia felis*, additional flea, tick, mite, and louse species have also been associated with this bacterium by molecular means; however, the role of these arthropods in the transmission of *R. felis* has not been clarified. Here, we succeeded in culture isolation of *R. felis* from a host-seeking castor bean tick, *Ixodes ricinus*, the most common tick in Slovakia. The bacterial isolation was performed on XTC-2 cells at 28 °C using the shell-vial technique. An evaluation of the growth properties was performed for both the XTC-2 and Vero cell lines. We observed *R. felis* in the infected host cells microscopically by Gimenez staining and immunofluorescence assay. The *R. felis* isolate was purified by gradient ultracentrifugation and visualized by electron microscopy. Fragments of the genes *glt*A, *omp*A, *omp*B, *htr*A, *rpo*B, *sca*4, *rff*E, and *rrs* were amplified and compared with the corresponding sequences of the type strain URRWXCal2 and other *R. felis* culture -isolated strains. We did not detect any nucleotide polymorphisms; however, plasmid pRFδ, characteristic of the standard strain, was absent in our isolate. Herein, we describe the first successful isolation and characterization of a tick-derived *R. felis* strain “Danube”, obtained from an *I. ricinus* nymph.

## 1. Introduction

The obligate intracellular bacterium, *Rickettsia felis,* is a vector-borne pathogen and the causative agent of flea-borne spotted fever [1]. Similar to other rickettsioses (e.g., murine typhus, rickettsialpox, and Mediterranean spotted fever), typical symptoms include fever, headache, maculopapular rash, fatigue, myalgia, arthralgia, elevation of liver enzymes, and eschar at the bite site [2,3]. The disease is considered clinically mild, but in older patients and in individuals with delayed diagnosis, it may cause severe illness with gastrointestinal and neurologic signs [4,5,6], including some cases of meningoencephalitis being fatal [7].

The distribution of *R. felis* appears to be cosmopolitan, as the organism has been detected in various countries worldwide, with an increasing number of human case reports [8,9,10]. In Eurasia, *R. felis* was detected for the first time in southern Spain in the cat flea *Ctenocephalides felis*, the biological vector and reservoir of this bacterium [11]. Although *R. felis* is primarily identified in the cat flea, it has also been associated with other arthropods [12,13]. By molecular techniques, *R. felis* has been detected in different flea species, mosquitoes, mites, and ticks (both Ixodid and Argasid). These tick species include *Rhipicephalus sanguineus* in Brazil, Spain, Chile, China, and the Philippines [14,15,16,17,18,19,20,21]; *Rhipicephalus bursa* in Turkey [22]; *Rhipicephalus turanicus* in Italy [23]; *Haemaphysalis flava, Haemaphysalis kitasatoe,* and *Ixodes ovatus* in Japan [24]; *Haemaphysalis sulcata* in Croatia [25]; *Haemaphysalis leporispalustris* in the United States [26]; *Haemaphysalis* sp. and *Rhipicephalus microplus* in Malaysia [27]; *Heamaphysalis bancrofti* in Australia [28]; *Amblyomma maculatum* in the United States [29]; *Amblyomma cajennense*, *Amblyomma humerale, Amblyomma ovale*, and *Amblyomma sculptum* in Brazil [14,30,31,32]; *Ixodes granulatus* in Taiwan [33]; *Ixodes hexagonus* in Italy [34]; *Ixodes ricinus* in Germany, France, Spain, Romania, and Serbia [35,36,37,38,39,40]; *Dermacentor nitens* in Brazil and Cuba [21,41]; and *Dermacentor variabilis* and *Carios capensis* in the United States [42,43] (Appendix A). The previous reports of the presence of *R. felis* in hard and soft ticks are not necessarily related to a vectorial role, and to date, the role of ticks in the ecology and transmission of *R. felis* has not been clarified. Its presence in ticks could be the result of an ingested blood meal from a bacteremic vertebrate host and a subsequent vertical transmission or from an infected flea via cofeeding transmission [12]. In Slovakia, the presence of *R. felis* was recently confirmed in *Ctenophthalmus solutus* fleas collected from *Apodemus agrarius* [44,45].

After the initial discovery of *R. felis* (historically known as the “ELB agent”) in cat flea tissues by electron microscopy [46], several attempts were made to establish a sustained bacterial culture, but that constituted a challenging task [47,48,49,50]. Finally, the propagation of *R. felis*, isolated from commercial fleas maintained by Flea Data, Inc. (Freeville, New York, NY, USA), was successful [51]. This isolate, named Marseille-URRWXCal2, is the reference strain of *R. felis* [52].

Seven *R. felis* culture isolates have been reported up to now (Table 1). The growth of *R. felis in vitro* has been accomplished using several different cell lines: amphibian cells (XTC-2) [51,53,54], tick cells (ISE6) [55,56], mosquito cells (Aa23, Sua5B, C6/36) [57,58,59], fly cells (S2) [60], and mammalian cells (Vero) with special medium supplementation [51,52,61]. A common characteristic of the effective propagation of *R. felis* is incubation at lower temperatures, i.e., ≤32 °C [52,60,62]. All culture isolates, except the most recent one from Australia and the unique LSU-Lb strain, were obtained from commercial/laboratory colony cat fleas or cat fleas collected from dogs. In addition, Stevenson and colleagues isolated *R. felis* from *Anomiopsyllus nudata* fleas (collected from *Neotoma albigula* in the United States), but unfortunately, the authors could not establish a sustained culture of rickettsiae [63].

In the current study, we describe the successful isolation and cultivation of *R. felis* from a questing *I. ricinus* nymph by the shell-vial technique. The isolate named “Danube” was established in the amphibian XTC-2 cells line, and the partial characterization of its biological properties by molecular biology techniques and microscopy are discussed here.

## 2. Materials and Methods

### 2.1. Collection of Ticks

For culture isolation, host-seeking *I. ricinus* nymphs were collected in the borough of Podunajské Biskupice (48.07° N, 17.12° E, altitude about 130 m), in the south-eastern part of the Bratislava region (Figure 1). The area lies in the geographic zone of the Danube lowland, a grassland with edge habitats forested by *Quercus, Carpinus, Populus*, and *Salix*, and characterized by a moderate climate. The collection area is influenced by anthropic rural environmental activities (hunting and farming). The Bratislava region is also characterized by a high level of biodiversity of wild vertebrate species. All ticks were collected by blanket dragging from vegetation, morphologically identified according to a standard taxonomic key [64], and stored at −80°C until further investigation. No specific permissions were required for questing tick collection in this location as the region is not a protected area. The field study did not involve any endangered or protected species.

### 2.2. Isolation of Rickettsia from Ticks

Isolation of rickettsiae was performed as previously described [65] with minor modifications. Briefly, nymphs were thawed, sterilized by immersion in iodinated alcohol for 10 min, rinsed with sterile water for 10 min, and dried on sterile filter paper under a laminar flow hood. Nymphs were longitudinally cut, one half of the tick was triturated in 1 mL of Leibovitz’s medium (L-15; Lonza, Basel, Switzerland), and the mixture was placed into a shell-vial containing a monolayer of amphibian epithelial XTC-2 cells [66] (the cell line was obtained from the laboratories of the Reference Center for Rickettsioses, Faculty of Medicine, Marseille, France). All ticks were processed individually. The shell-vials were centrifuged at 700× *g* for 20 min, and the supernatant was replaced with 1 mL of supplemented Leibovitz’s medium containing 2.5% fetal bovine serum (FBS; HyClone, Logan, UT, USA), 8% tryptose phosphate broth (TPB; Sigma-Aldrich, St. Louis, MO, USA), and 2 mM L-glutamine (Lonza). After 7 days of incubation at 28 °C, scraped XTC-2 cells were applied to a microscope slide and stained by the method of Gimenez [67] to monitor the infection status of host cells by rickettsiae. Supernatant with detached cells from the infected shell-vials was transferred into cell culture flasks with a growth area of 25 cm^2^ (Corning Inc., Corning, NY, USA). Fresh supplemented Leibovitz’s medium was added, and the flasks were incubated at 28 °C in the absence of CO_2_.

### 2.3. Cultivation and Purification of Isolated R. felis

The bacteria were sub-cultured without trypsinization in the XTC-2 cell line with supplemented Leibovitz’s medium containing 2.5% FBS, 8% TPB, and 2 mM L-glutamine at 28 °C, and subsequently in the Vero cell line (ATCC CCL-81) with supplemented RPMI 1640 medium containing 25 mM HEPES, 2 mM L-glutamine, and 2.5% FBS at 32 °C and 34 °C in a 5% CO_2_ atmosphere. Host cells were examined daily for cytopathic effect formation due to rickettsial infection, and the medium was replaced weekly to ensure adequate nutrition. When cells showed infection with apparent cytopathic effect (≥90%), aliquots were used to establish new infections in XTC-2 cell monolayers or were frozen at −80 °C, and stored for future reference. The presence of *R. felis* in the cell culture was verified by Gimenez staining using an Eclipse Ni-U microscope (Nikon, Tokyo, Japan), and photographs of stained slides were taken using a Coolpix P330 camera (Nikon, Tokyo, Japan). Only contrast was adjusted on the images.

The propagated isolate of *R. felis* in the XTC-2 cell line was purified by a modified isopycnic density gradient centrifugation protocol [68,69]. Briefly, infected cells were harvested from flasks and pelleted by centrifugation at 1000× *g* for 5 min at 4 °C. The cell culture medium was removed, and cold K36 buffer (0.1 M KCl, 0.015 M NaCl, 0.05 M potassium phosphate buffer, pH 7.0) was added to each sample. Host cell lysis was achieved by sonication, and rickettsiae were separated from host cell debris by centrifugation at 1000× *g* for 5 min at 4 °C (Hettich, Tuttlingen, Germany). Supernatants containing bacteria were transferred to new tubes and centrifuged at 22,000× *g* for 30 min at 4 °C. Pelleted rickettsiae were resuspended in SPG buffer (0.218 M sucrose, 3.76 mM KH_2_PO_4_, 7.1 mM K_2_HPO_4,_ and 4.9 mM potassium glutamate) and overlaid onto a discontinuous Renografin gradient (Ultravist 370, Bayer, Leverkusen, Germany) made with three concentrations: 42%, 36%, and 30% solution. Sample tubes were then centrifuged at 120,000× *g* for 90 min at 4 °C. After ultracentrifugation (Appendix A), viable rickettsiae were collected into clean tubes and centrifuged at 22,000× *g* for 30 min at 4 °C. The bacterial pellet was resuspended in cold SPG buffer and stored at −80 °C for further experiments.

### 2.4. Monitoring of Rickettsial Intracellular Growth

The cell density of viable XTC-2 cells was enumerated using trypan blue staining in a Bürker Counting Chamber, and afterward, cells were seeded into 12-well tissue culture plates (Greiner Bio-One, Kremsmünster, Austria). The monolayers of host cells were infected with *R. felis* at a multiplicity of infection (MOI) equal to 10. The infection was initiated from the frozen stocks of purified rickettsiae stored in SPG buffer. The concentration of bacteria was determined as described previously [70]. On each day post-infection (dpi) up to one week, infected XTC-2 cells were scraped into culture media and centrifugated at 16,000× *g* for 15 min at 4 °C. The supernatants were discarded, and the pelleted samples were stored at −80 °C until further nucleic acid extraction. Two separate biological replicates were carried out for each collection time point.

### 2.5. RNA Extraction and Reverse-Transcriptase Quantitative PCR (RT-qPCR)

Total RNA from infected XTC-2 cells was extracted using the RNeasy Mini Kit (Qiagen, Hilden, Germany) according to the manufacturer’s instructions. Extracted RNA was treated with RNase-Free DNase Set (Qiagen) and purified using the RNeasy MinElute Cleanup Kit (Qiagen). The quality and quantity of the extracted RNA samples were evaluated with a nanophotometer (Implen, Westlake Village, CA, USA), and the subsequent reverse transcription was performed on 1 µg of isolated RNA using the First Strand cDNA Synthesis Kit with random hexamer primers (Thermo Scientific, Waltham, MA, USA). No-reverse transcriptase controls (reverse transcription without the enzyme) and negative controls (ultrapure water instead of a template) were included for cDNA synthesis and RT-qPCR. All samples were stored at −20 °C until further processing.

To determine rickettsial transcript copy numbers of the *rps*L gene (encoding ribosomal S12 protein), we performed RT-qPCR as published previously [70]. The PCR reactions were carried out in a CFX 96 Real-Time system C 1000 Thermal Cycler (Bio-Rad, Hercules, CA, USA). The components of the PCR mixture included Maxima Probe/ROX qPCR Master Mix (Thermo Scientific, Waltham, MA, USA), 0.4 µM of each primer, 0.4 µM of TaqMan probe, ultrapure nuclease-free water, and 12.5 ng of cDNA template, or serial dilution of standards. The amplification conditions were as follows: an initial denaturation step at 95 °C for 10 min, followed by 40 cycles of denaturation at 95 °C for 20 s, annealing, and elongation at 58 °C for 40 s, with fluorescence acquisition in single mode. Each sample was tested in technical triplicate. To generate a standard curve, we utilized a 10-fold serial dilution of the amplified and purified PCR product of the gene *rps*L from *Rickettsia akari* strain MK (Kaplan, ATCC VR-148) [71].

### 2.6. DNA Extraction

Total genomic DNA was extracted from the remaining half-ticks and rickettsial culture samples using the QIAamp DNA Mini kit (Qiagen, Hilden, Germany) according to the manufacturer’s instructions. Plasmid DNA from purified *R. felis* was extracted with QIAprep Spin Miniprep Kit (Qiagen, Hilden, Germany) following the manufacturer’s instructions. Extracted DNA samples were eluted in ultrapure nuclease-free water (Thermo Scientific, Waltham, MA, USA). The quality and quantity of DNA were evaluated using a nanophotometer (Implen, Westlake Village, CA, USA), and the samples were stored at −20 °C until further use. To avoid contamination, DNA extraction, PCR setup, and agarose gel analysis were carried out in separate rooms.

### 2.7. Molecular Identification by PCR and Sequencing

The samples from the nymphs, infected cell lines, and purified bacteria were tested by conventional PCR carried out in a GenePro Thermal Cycler (Bioer Technology, Hangzhou, China) and/or Labcycler (Sensoquest, Göttingen, Germany). Primer sets amplifying fragments of the rickettsial genes *rrs* [72,73], *sca*4 [74], *htr*A [24], *glt*A [51,75,76], *omp*A [55,77], *omp*B [78], *rpo*B [52,79], and *rff*E [71] are listed in Appendix A. The PCR mixture included FIREPol Master Mix Ready to Load (Solis BioDyne), 0.25 µM of each primer, ultrapure nuclease-free water, and 100 ng of extracted DNA template. The thermocycling parameters were those published previously (Appendix A). The culture was also checked for any *R. helvetica* contamination using species-specific primers (rhelv.26f and rhelv.356r) targeting the 23S rRNA coding gene [80]. Positive controls (genomic DNA from *Rickettsia slovaca* strain B-13, ATCC VR-1639 and/or *Rickettsia helvetica* strain C9P9, ATCC VR-1375), and no template controls were included in all PCRs.

To amplify a portion of the *R. felis* pRF and pRFδ plasmid sequences, we used oligonucleotides published previously [81]. The PCR reaction was composed of Platinum *Taq* DNA High Fidelity Polymerase (1 U per reaction; Invitrogen, Waltham, MA, USA), High Fidelity PCR Buffer, 2.0 mM MgSO_4_, 0.2 mM of each dNTP, 0.4 µM of each primer, ultrapure nuclease-free water, and 2 µL of template DNA.

The amplified PCR products were separated by electrophoresis in 1% agarose gels and visualized with SYBR Safe DNA gel stain (Invitrogen, Waltham, MA, USA) using a Benchtop 3UV Transilluminator (UVP, Upland, CA, USA). The DNA fragments were cleaned up with ExoSAP-IT (Thermo Scientific, Waltham, MA, USA) and sequenced at least twice, in both forward and reverse directions (Microsynth Seqlab, Göttingen, Germany). The obtained sequences were aligned with BioEdit Sequence Alignment Editor software [82] and compared with those of *Rickettsia* spp. available in GenBank by means of the Basic Local Alignment Search Tool (http://www.ncbi.nlm.nih.gov/BLAST/, 24 March 2022).

### 2.8. Immunofluorescence Assay (IFA)

Immunofluorescence detection of *R. felis* in the infected host cells was performed as previously described [83]. Confluent XTC-2 and Vero cells were infected with an MOI 25. After 72 h, host cells were fixed with 4% paraformaldehyde in cytoskeleton-stabilizing PHEM buffer [84], permeabilized with 0.1% Triton X-100 in PBS, and blocked with 5% milk in PBS. Cells were then incubated with a rabbit polyclonal primary antibody against *Rickettsia conorii* (1:100; prepared in our experimental animal facility at the Biomedical Research Center, Slovak Academy of Sciences, Bratislava, Slovakia). For the localization of intracellular bacteria, we applied a goat anti-rabbit IgG (H + L) polyclonal secondary antibody conjugated with Rhodamine Red-X (Invitrogen, reference number R6394, lot 1402199, made in the USA). To visualize filamentous actin, coverslips were probed with Alexa Fluor 488-tagged phalloidin (Invitrogen, reference number A12379, lot 1378369, made in the United States). Finally, samples were mounted with Vectashield Medium (Vector Laboratories, Burlingame, CA, USA) containing DAPI and analyzed using a Zeiss LSM 510 META confocal fluorescent microscope. Brightness and contrast were adjusted by LSM Image browser software (Zeiss, Jena, Germany).

### 2.9. Electron Microscopy

For negative staining, 5 μL of the sample was placed onto glow-discharge activated [85] carbon/formvar grids. After 30 s of adsorption, the grids were negatively stained with 1% ammonium molybdate and 0.1% trehalose for 30 s. The grids were air-dried and examined in an FEI/Philips CM100 electron microscope (FEI).

For scanning electron microscopy, the purified samples of *R. felis* isolate were fixed with 3% glutaraldehyde in SP buffer (3.76 mM KH_2_PO_4_, 7.1 mM K_2_HPO_4_, 0.218 M sucrose) overnight at 4 °C. The washed cells were then allowed to sediment overnight onto poly-l-lysine-treated circular coverslips or silicon wafers at 4 °C. The coverslips and the wafers with attached bacteria were dehydrated through an alcohol series and critical point dried from liquid CO_2_ in a K850 Critical Point Dryer (Quorum Technologies, Lewes, United Kingdom). The dried samples were sputter-coated with 3 nm of platinum in a Q150T Turbo-Pumped Sputter Coater (Quorum Technologies, Lewes, United Kingdom). The final samples were examined in an FEI Nova NanoSEM scanning electron microscope (FEI) at 5 kV using CBS and TLD detectors.

### 2.10. Statistical Analysis

For evaluation of rickettsial growth in host cells, one-way analysis of variance (ANOVA) was performed, followed by Dunnett’s multiple comparison post-hoc test when an overall significance (*p*-value of <0.05) was found. Analyses were performed using Prism 9 software (GraphPad Software, San Diego, CA, USA).

## 3. Results and Discussion

### 3.1. Initial Isolation of R. felis from an I. ricinus Nymph

Sixty unfed *I. ricinus* nymphs, collected from vegetation by flagging, were processed in the experiment. After surface sterilization, nymphs were longitudinally cut in half, and one half was used for culture isolation and the second half for an initial molecular screening. The presence of rickettsiae was confirmed in 20% of samples by conventional PCR, using the primers CS877p and CS1258n. In parallel, we focused on a positive culture isolate that we recovered from a tested *I. ricinus* nymph. By Gimenez staining, we detected intracellular, rod-shaped bacteria in the host cells cultured in the shell-vial 7 days after inoculation. Following an apparent cytopathic effect in XTC-2 cells three weeks after inoculation, we harvested and sub-cultured the infected cells into a confluent monolayer of XTC-2 cells, first in 25 cm^2^ and later in 75 cm^2^ tissue culture flasks to establish a sustained bacterial culture. In order to identify the species of the isolated rickettsiae, we tested samples from infected XTC-2 cells by PCR using genus-specific primers, amplifying the fragments of the *glt*A and *rff*E genes, coding the citrate synthase and UDP-N-acetylglucosamine 2-epimerase, respectively (Figure 2). Next, the amplification products were sequenced (#ON053296, #ON053294), and the isolated rickettsiae were confirmed as *R. felis*, with 100% identity to the previously reported gene fragment sequences of the type strain URRWXCal2 and significant nucleotide differences from other rickettsiae. The isolate was named the “Danube” strain (for the tick collection site in the Danube lowland).

The use of the XTC-2 cell line has proven effective in the isolation of other *R. felis* strains, including the type strain and the culture isolates reported from Australia (Table 1). Our laboratory successfully employed the shell-vial technique previously to isolate other intracellular microorganisms from arthropods, for instance, *Rickettsia monacensis* IRS3 [86], *R. slovaca* [87], *Diplorickettsia massiliensis* [88], *R. helvetica* [65], *Arsenophonus nasoniae* [89], and *Rickettsia raoultii* [90].

For further studies, we purified the rickettsiae collected from the infected XTC-2 cells by renografin density gradient centrifugation (Appendix A). As *R. helvetica* is the most prevalent rickettsia in *I. ricinus* ticks in Slovakia [91,92,93,94], we performed species-specific amplification of the 23S rRNA gene to confirm the absence of *R. helvetica* in the cultured *R. felis* isolate Danube (Appendix A). *Rickettsia helvetica* could not be detected in the purified sample of *R. felis* by PCR, whereas the corresponding primer pair amplified the positive control DNA.

### 3.2. Propagation of Rickettsiae in XTC-2 and Vero Cell Lines

The *R. felis* isolate adapted well to propagation in an XTC-2 cell-containing culture flask after sub-culturing from the shell-vial. We observed an evident cytopathic effect in infected host cells (similar to the standard strain URRWXCal2 [52]), and the bacterial proliferation was also confirmed by Gimenez staining (Figure 3). Next, we attempted to infect the mammalian Vero cells with our *R. felis* isolate at 32 °C and 34 °C. *Rickettsia felis* Danube was able to infect Vero cells, inducing round cytopathic foci at 32 °C, similar to those of the infected XTC-2 cell monolayers (Figure 3). Comparably to the reference strain [51], *R. felis* Danube replicated better in XTC-2 cells than in the mammalian cell line. In Vero cells maintained at 34 °C, the number of propagating bacteria gradually decreased. A similar observation was reported during the early cultivation of the “ELB” agent in Vero cells at 34 °C [47].

To extend the knowledge regarding the growth characteristics of *R. felis* Danube in XTC-2 cells, we propagated the bacteria in 12-well plates to estimate the multiplication rate. Samples collected every 24 h after infection were subjected to qPCR, and the mean number of rickettsial genome equivalents was calculated for each well, based on the amplification of the rickettsial single-copy, house-keeping gene *rps*L compared to the standard calibration curve. As displayed in Figure 4a, *R. felis* proliferated rapidly in host cells, with a statistically significant increase in the mean value of the bacterial genome equivalents over the time course of our experiment. The growth curve showed an initial lag phase (1–2 dpi), followed by an exponential increase in *R. felis* genome copy numbers (3–6 dpi). We observed the highest multiplication rate at 6 dpi, followed by a stationary phase.

To demonstrate the viability of rickettsiae in the infected XTC-2 cells, we extracted RNA from the parallel cultivation wells and quantified the synthesized complementary DNA by qPCR, in the same manner as the extracted DNA samples. Similar to the rickettsial growth kinetics based on the estimated numbers of genomic equivalents, we observed an increase in the mean transcript numbers of the *R. felis rps*L gene (Figure 4b). Following the highest transcript number of the ribosomal gene at 5 dpi, we observed a decrease in rickettsial cDNA quantity, which corresponds to the heavy host cell detachment due to bacterial burden observed by light microscopy (data not shown).

### 3.3. Molecular Characterization of the R. felis Isolate Danube

In addition to confirming the species identity and assessing the nucleotide variability of conserved rickettsial genes in *R. felis* Danube, fragments of the *omp*A and *omp*B (outer membrane proteins A and B), *sca*4 (surface cell antigen 4), *glt*A, *htr*A (17 kDa surface antigen), *rpo*B (RNA polymerase subunit β), and *rrs* (16S rRNA) genes were amplified by conventional PCR, including *R. felis*-specific primer sets [55,77] (Figure 5a). The subsequent sequence analysis confirmed the isolate Danube as *R. felis*, with 99–100% similarity to those sequences obtained from other culture-isolated strains of *R. felis* (Appendix A). Partial sequences of *R. felis* Danube generated in this study were deposited into GenBank and assigned accession numbers ON053297, ON053298 (*glt*A), ON053299, ON053306 (*omp*A), ON053301-ON053303 (*omp*B), ON053304 (*sca*4), ON053294 (*htr*A), ON053305 (*rpo*B), and ON053300 (*rrs*).

The complete genome of the *R. felis* strain URRWXCal2 was previously described, and unique characteristics, including the presence of two conjugative plasmids, pRF and pRFδ, were revealed [95]. Recently, additional strains of *R. felis*, LSU and LSU-Lb, were subjected to genome and plasmid content analysis [96]. In our study, we obtained positive results for the presence of the pRF plasmid using the primer sets pRFa and pRFb, and pRFc and pRFd (Figure 5b), and the sequences amplified in the current study (#ON053307) were identical to those reported for *R. felis* previously. However, we could not detect the pRFδ plasmid sequence by PCR (primers pRFa and pRFd) in any of our *R. felis* Danube samples. Similar findings were described for the *R. felis* strains LSU and LSU-Lb [55,56], and the *R. felis* genotype RF2125 endemic in *Archaeopsylla erinacei* fleas from Algeria [97], confirming plasmid content variation between different *R. felis* strains [81].

### 3.4. Microscopy

*Rickettsia felis* Danube was observed in infected XTC-2 and Vero cells by indirect immunofluorescence assay. To visualize rickettsiae, we applied anti-*R. conorii* rabbit serum as the primary antibody, followed by a rhodamine-conjugated secondary antibody. Serologically, spotted fever group rickettsia antibodies are cross-reactive with *R. felis* antigens [98,99]. We labeled the actin filaments of host cells with conjugated phalloidin, and cell nuclei were counterstained with DAPI. Confocal microscopy demonstrated that rickettsiae were present in the cytoplasm of the infected host cells (Figure 6), similar to the initial isolate “ELB” and the reference strain [49,51]. Unlike the standard strain, but similar to *R. felis* LSU, we did not observe evident intracellular actin polymerization by IFA [55].

To visualize the ultrastructure of *R. felis* Danube, we examined the purified rickettsiae using transmission and scanning electron microscopy (Figure 7). The morphology of the isolate (appearing as a typical-shaped coccobacillus) was characteristic of rickettsiae. The size of the detected microorganisms was 0.3 µm in width and 0.9 µm in length. We repeatedly observed bacteria undergoing binary fission. Even after comprehensive purification, occasional host cell debris could be seen in our samples.

### 3.5. A Possible Role of Ticks in the Ecology of R. felis

The use of molecular biology and cell culture techniques to uncover microorganisms in arthropod disease vectors has enabled new discoveries. The list of circulating pathogens that can be found in Slovakia has broadly expanded in the past decade. Although successful cultivation of *R. felis* has been achieved before (Table 1), this is the first cultivation of a tick-derived strain, and the first isolate originated from Europe. Our study revealed that ticks in Slovakia, alongside fleas [44,45], can harbor viable, infectious *R. felis*.

In the present study, *R. felis* was isolated from a questing *I. ricinus* nymph, collected from vegetation in spring. We assume that a larval tick was most likely infected the year before in summer, molted, and then overwintered with the rickettsial pathogen. The dormancy phenomenon (diapause) in Central Europe is part of the *I. ricinus* life cycle [100]. Similar to our finding, *R. felis* was identified in a questing *I. ricinus* nymph in France [36], indicating transstadial transmission of the bacteria.

The acquisition of rickettsiae by ticks occurs via feeding on a rickettsiaemic vertebrate host or by cofeeding transmission between arthropods. Small rodents, insectivores, and birds are important hosts for the immature stages of Ixodid ticks, including the castor bean tick [101]. *Rickettsia felis* DNA has been detected in ear tissues of *Apodemus sylvaticus, Apodemus flavicollis,* and *Myodes glareolus* in Germany [102,103]. In addition, hedgehogs can carry *R. felis*-positive ticks and fleas and may act as a reservoir host for pathogens [104,105,106]. Recently, *R. felis* was also identified in a *I. ricinus* nymph removed from a common blackbird (*Turdus merula*) in Romania [39].

Immature ticks can compensate for the absence of birds and small mammals by feeding on larger animals [101]. Feral and/or companion cats and dogs; domesticated animals such as cattle, horses, and sheep; as well as wildlife (all present in the tick collection area in Slovakia) can be reservoirs of vector-borne zoonotic *Rickettsia* spp., such as *R. felis* (Appendix A), and may facilitate the horizontal transmission of these pathogens.

Ticks can harbor a number of bacterial species within the *Rickettsia* genus and are capable of transmitting these microorganisms to humans during a bite. Lejal and colleagues detected *R. felis* in the salivary glands of unfed, adult *I. ricinus* ticks by microfluidic real-time PCR [37]. A new publication from Serbia examining ticks feeding on humans and their blood samples revealed a case study of *R. felis* infection in an elderly patient parasitized by an adult *I. ricinus* female tick [40]. However, to date, culture isolation of *R. felis* from a clinical sample has not been achieved [3].

Much work still lies ahead to fully understand the complex ecology of *R. felis*. Laboratory experiments are necessary to clarify the vector competence of ticks in the transmission route of *R. felis*. Further investigation is needed regarding transovarial [107] and transstadial transmission of this microorganism to subsequent life stages of ticks. Also, additional evidence is required to verify the potential role of ticks in the transmission of *R. felis* to vertebrate hosts, including humans.

## 4. Conclusions

The present study described the successful isolation of a tick-derived *R. felis* strain obtained from an unfed *I. ricinus* nymph. The isolation was performed in XTC-2 cells at 28 °C by the shell-vial technique. Evaluation of bacterial growth was conducted for both XTC-2 and Vero cell lines. We observed *R. felis* in infected host cells by Gimenez staining and IFA. The new *R. felis* isolate was purified by gradient ultracentrifugation and visualized by electron microscopy. For genetic analysis, fragments of the genes *glt*A, *omp*A, *omp*B, *htr*A, *rpo*B, *sca*4, *rrf*E, and *rrs* were sequenced and compared with the corresponding sequences of the *R. felis* type strain URRWXCal2 and the other reported culture isolates. We did not detect any nucleotide variability; however, the plasmid pRFδ was not present in our isolate. Our findings demonstrated that ticks harbor viable *R. felis* in Slovakia. Further studies regarding the role of ticks in the ecology of *R. felis* are warranted.

## Figures and Tables

**Figure 1 ijerph-19-04321-f001:**
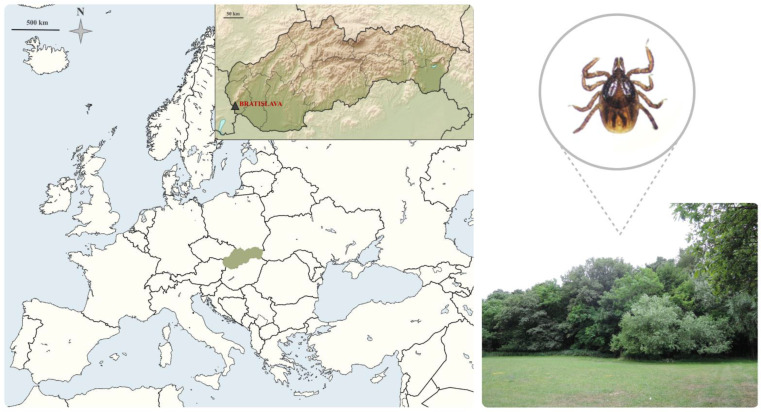
Tick sampling location in Slovakia. On the left: a political map of Europe indicating the location of Slovakia (in light green). Ticks were collected in the Bratislava district (marked with a triangle) by flagging method (maps from www.vidiani.com, accessed on 27 January 2022, licensed under CC-BY 3.0, desaturated from the original). On the right: *I. ricinus* nymph collected from the local vegetation (SMZ1500 Stereomicroscope).

**Figure 2 ijerph-19-04321-f002:**
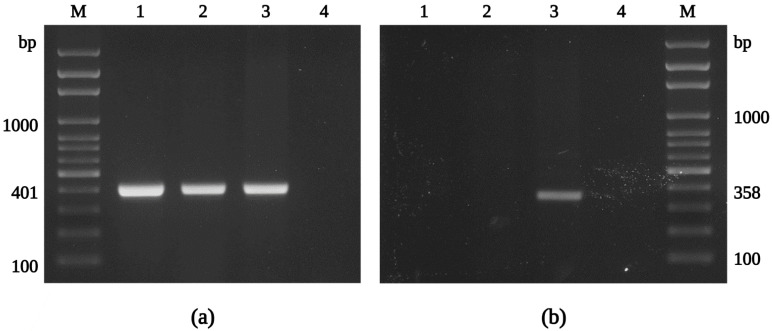
Identification of the isolate Danube by PCR. Agarose gel electrophoresis of PCR products of the genes *glt*A (**a**) and *rff*E (**b**), amplified with primers CS-78 and CS-323, MQ32 and MQ33, respectively. M: molecular marker; 1: *R. helvetica* C9P9; 2: *R. helvetica* IR16; 3: *R. felis* Danube; 4: negative control. Note: primer set MQ32, MQ33 does not amplify *R. helvetica*.

**Figure 3 ijerph-19-04321-f003:**
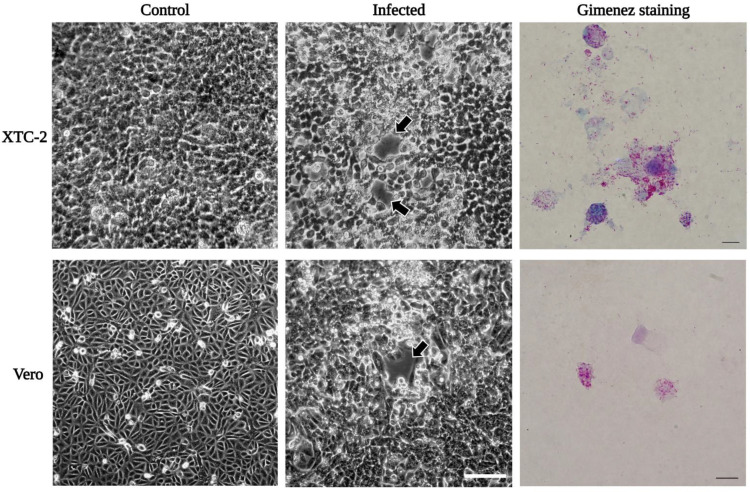
Growth of *R. felis* Danube in XTC-2 and Vero cells. Rickettsiae were propagated in XTC-2 and Vero cells at 28 °C and 32 °C, respectively. We observed an apparent cytopathic effect (plaque formation, black arrows) in the infected host cells at 6 dpi in XTC-2 and 14 dpi in Vero cells (Zeiss Axiovert 40 CFL trinocular inverted phase-contrast microscope; scale bar: 100 µm). *Rickettsia felis* stained magenta in contrast to host cells in blue by Gimenez technique (Leica DM 4500B microscope; scale bar 10 µm).

**Figure 4 ijerph-19-04321-f004:**
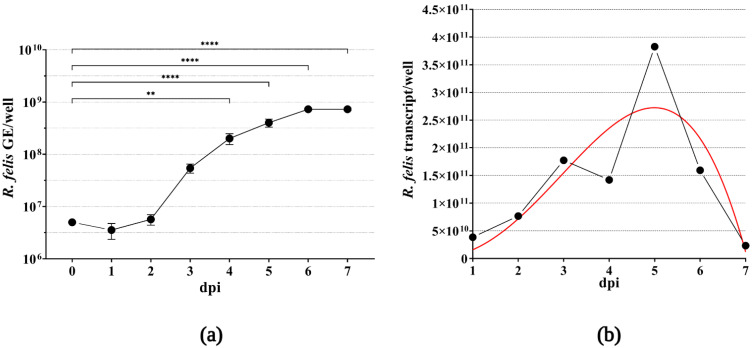
Graphical representation of the rickettsial rate of intracellular growth in XTC-2 cells. Amphibian host cells incubated in 12-well plates and approaching confluence were infected with MOI 10 rickettsiae. Infected cells from culture wells were harvested in 24 h intervals. To demonstrate the growth kinetics of *R. felis* Danube, the approximate mean numbers of bacterial genome equivalents (GE) were calculated based on the amplification of the *rps*L gene by qPCR. Standard errors of the mean were calculated from two biological replicates. For statistical analysis, one-way ANOVA was performed, followed by Dunnett’s multiple comparison test (** *p* < 0.01, **** *p* < 0.0001) (**a**). To confirm viability of rickettsiae, the estimated transcript numbers of the same gene were evaluated per well, as described in Materials and Methods. The red line represents nonlinear regression, Beta growth then decay (**b**).

**Figure 5 ijerph-19-04321-f005:**
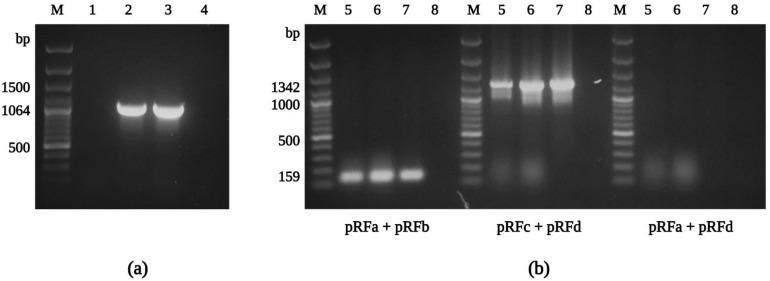
Agarose gel electrophoresis of PCR amplicons of the *omp*A gene and plasmids from *R. felis* Danube. The partial sequence of the gene *omp*A was amplified using *R. felis*-specific primers Rf190.1790fw and Rf190.2857rev (**a**). The pRF plasmid was detected using the primer pairs pRFa-pRFb (expected size 159 bp) and pRFc-pRFd (expected size 1342 bp), but not pRFδ (primers pRFa and pRFd; expected size 1168 bp) (**b**). M: molecular marker; 1: uninfected XTC-2 cells; 2: infected XTC-2 cells; 3: purified *R. felis* Danube; 4: negative control; 5: infected XTC-2 cells; 6: purified *R. felis* Danube; 7: extracted plasmid DNA from purified *R. felis* Danube; 8: negative control.

**Figure 6 ijerph-19-04321-f006:**
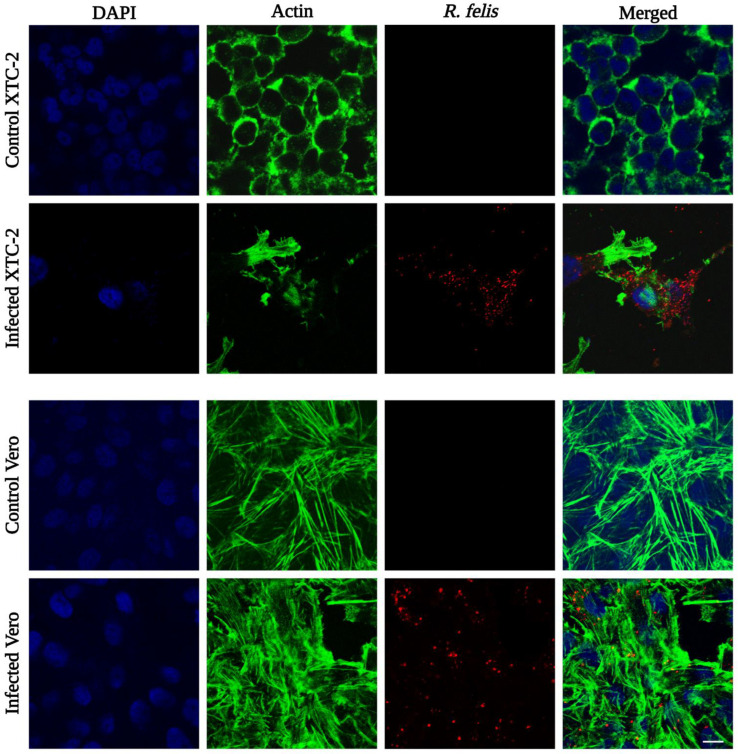
Detection of *R. felis* Danube by immunofluorescence assay. Infected XTC-2 and Vero cells were fixed with 4% paraformaldehyde in PHEM buffer, and rickettsiae were labeled with rabbit polyclonal antibody against *R. conorii*, followed by rhodamine-conjugated goat anti-rabbit antibody (red signal). Actin filaments of host cells were labeled with conjugated phalloidin (green signal), and cell nuclei were stained with DAPI (blue signal). Uninfected cells treated with both primary and secondary antibodies were used as negative controls (scale bar 10 µm).

**Figure 7 ijerph-19-04321-f007:**
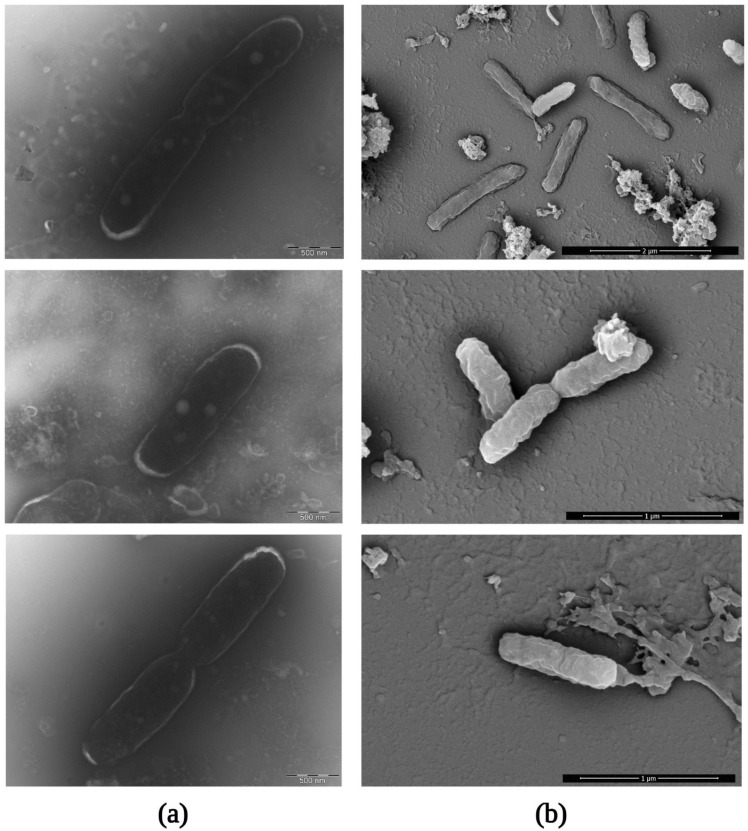
Visualization of purified *R. felis* Danube by electron microscopy. Negative staining of rickettsial binary fission captured by transmission electron microscopy (**a**). Scanning electron microscopy of purified and glutaraldehyde-fixed *R. felis* sedimented onto circular coverslips and silicon wafer (**b**).

**Table 1 ijerph-19-04321-t001:** *Rickettsia felis* culture-isolated strains.

Strain	Year	Origin	Cell Culture	Reference
URRWXCal2	2001	Cat flea of the colony maintained by Flea Data, Inc., in the United States	*Xenopus laevis* cell line XTC-2 (28 °C)	Raoult D. et al. *Emerg Infect Dis* 7(1): 73–81 [51]
Pedreira	2006	Cat flea collected from a naturally infested farm dog, in Brazil	*Aedes albopictus* cell line C6/36 (25 °C)	Horta M.C. et al. *Appl Environ Microbiol* 72(2): 1705–1707 [57]
LSU	2006	Cat flea of the laboratory colony maintained at the Louisiana State University, in the United States	*Ixodes scapularis* cell line ISE6 (32 °C)	Pornwiroon W. et al. *Appl Environ Microbiol* 72(8): 5589–5595 [55]
LSU-Lb	2011	Booklouse *Liposcelis bostrychophila* from shredded corn cobs, in the United States	*Ixodes scapularis* cell line ISE6 (32 °C)	Thepparit C. et al. *PLoS ONE* 6(1): e16396 [56]
CfCR(SJ)	2011	Cat flea collected from a domestic dog, in Costa Rica	*Aedes albopictus* cell line C6/36 (28 °C)	Hun L. et al. *Vector Borne Zoonotic Dis* 11(10): 1395–1397 [58]
N/A *	2013	Cat flea collected from a pound dog, and the flea laboratory colony maintained at the University of Queensland, in Australia	*Xenopus laevis* cell line XTC-2 (28 °C)	Hii S.F. et al. *Parasit Vectors* 6:159 [53]
N/A *	2020	Domestic dog infected with *R. felis* in a laboratory setting, in Australia	*Xenopus laevis* cell line XTC-2 (28 °C)	Ng-Nguyen D. et al. *Sci Rep* 10: 4151 [54]
Danube	This study	Questing *Ixodes ricinus* nymph, in Slovakia	*Xenopus laevis* cell line XTC-2 (28 °C)	This study

* N/A—not available.

## Data Availability

The datasets used and analyzed during the current study are available from the corresponding author upon reasonable request.

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
