# Peer review of "Culture Isolate of Rickettsia felis from a Tick"

_ijerph, 2022, doi:10.3390/ijerph19074321_

Round 1
Reviewer 1 Report
This is a very interesting piece of work, which is well presented and novel. My comments are generally stylistic but there are some more important methodological queries to answer.
Important points to clarify in methods and results:
- How many culture attempts from ticks were attempted and had R. felis already been detected by molecular methods in some of the nymphs before isolation attempts?
- How long was the period before rickettsiae became visible in initial cultures?
- For figure 4, how many replicates were used at each time-point and were these biological replicates (separate culture wells) or technical replicates (same sample with replicates on the qPCR plates)? If the latter, the statistical analysis is not appropriate. Sample sizes should be given in the legend.
- In figure 6 (confocal), the resolution/magnification is not sufficient to claim that rickettsiae are present in host cell nuclei. Z-stack images of nuclei or TEM evidence should be provided for claim, or otherwise the claim must be heavily qualified.
Stylistic points:
- The combined Results and Discussion is an unconventional format for this sort of study. Please clearly separate the two sections.
- The Discussion is too long for a small study like this and reads more like a literature review towards the end (there are 125 citations, which reinforces this point). The co-feeding hypothesis is speculative and not really relevant to the study findings. After line 417, the Discussion increasingly loses focus and strays further from the study topic into the global epidemiology of R. felis. These final paragraphs should be removed.
- Table 1 is quite useful but very long; I suggest it is moved into the supplement. Table 3 is not very informative; the lack of polymorphisms in the Danube isolate is already explained in the text. For completeness, it could be retained in the supplement but would be better formatted in landscape to prevent line breaks. "*Six primer combined" doesn’t seem to refer to anything in the table.
- The pers. comm. from Kevin Macaluso seems unnecessary as it not disclosing an unpublished result. Written evidence of permission to use pers. comms. should always be provided in any case.
- The Institutional Review Board Statement should be removed as no vertebrate animals were used in the study.
- Although generally well written, there are a number of typographical and linguistic errors in the manuscript and some incomplete references. These errors are corrected in the attached PDF.

Reviewer 2 Report
This study describes the isolation and cultivation of “Dan- 78 ube” R. felis isolate from I. ricinus nimph in the amphibian XTC-2 cells line, and the partial characterization of its biological properties by molecular biology techniques and microscopy. The work is well written and structured in all sections and it deserves to be published in the present form.
Only few points should be modified:
L 27: please, detail the species of rickettsia that give similar symptoms
L 37 R. felis at this point in italics and throughout the text
L38-39 you should move this sentence after the description of tick involved in the R. felis infection
L 90 morphologically
L146-147 please rephrase this sentence. It lacks clarity
L 191 indicate the reference
Reviewer 3 Report
This is a good experiment, the authors isolated, cultured and characterized the R. felis from Ixodes ricinus, and overall well written. But there are still several issues that need to be improved.
- In Figure 2, the authors determined that the isolate was R. felis by PCR, but there is a problem that although the gene (rffE gene) is specific for R. felis, it is not enough to identify with just one gene in this key experiment. There should be other specific genes, right? The authors are invited to consider whether to supplement at least three genes to confirm the isolated pathogen.
- In Figure 3, because the isolates in this manuscript were discovered for the first time, if possible, the authors were asked to add the control pathogens (reference strain) to the cells to compare with the current isolation. In general, the authors wish to add more controls to compare with this isolate in each experiment.
Round 2
Reviewer 3 Report
Thank you for the author's response. It is recommended to accept and publish in the present form.
Author Response
Dear Reviewer # 3,
As You wrote:
Comments and Suggestions for Authors
Thank you for the author's response. It is recommended to accept and publish in the present form.
Submission Date
28 January 2022
All we can say:
Thank You very much for your recommendation to accept and publish the manuscript in the present form.
Zuzana Sekeyová,
Corresponding author